# Maternal cortisol is associated with neonatal amygdala microstructure and connectivity in a sexually dimorphic manner

David Q Stoye[1], Manuel Blesa[1], Gemma Sullivan[1], Paola Galdi[1], Gillian J Lamb[1], Gill S Black[1], Alan J Quigley[2], Michael J Thrippleton[3], Mark E Bastin[3], Rebecca M Reynolds[4†], James P Boardman[1,3†*]

[1]MRC Centre for Reproductive Health, University of Edinburgh, Edinburgh, United Kingdom; [2]Department of Radiology, Royal Hospital for Sick Children, Edinburgh, United Kingdom; [3]Centre for Clinical Brain Sciences, University of Edinburgh, Edinburgh, United Kingdom; [4]Centre for Cardiovascular Science, University of Edinburgh, Edinburgh, United Kingdom

**Abstract** The mechanisms linking maternal stress in pregnancy with infant neurodevelopment in a sexually dimorphic manner are poorly understood. We tested the hypothesis that maternal hypothalamic-pituitary-adrenal axis activity, measured by hair cortisol concentration (HCC), is associated with microstructure, structural connectivity, and volume of the infant amygdala. In 78 mother-infant dyads, maternal hair was sampled postnatally, and infants underwent magnetic resonance imaging at term-equivalent age. We found a relationship between maternal HCC and amygdala development that differed according to infant sex. Higher HCC was associated with higher left amygdala fractional anisotropy ($\beta$ = 0.677, p=0.010), lower left amygdala orientation dispersion index ($\beta$ = −0.597, p=0.034), and higher fractional anisotropy in connections between the right amygdala and putamen ($\beta$ = 0.475, p=0.007) in girls compared to boys. Furthermore, altered amygdala microstructure was only observed in boys, with connectivity changes restricted to girls. Maternal cortisol during pregnancy is related to newborn amygdala architecture and connectivity in a sexually dimorphic manner. Given the fundamental role of the amygdala in the emergence of emotion regulation, these findings offer new insights into mechanisms linking maternal health with neuropsychiatric outcomes of children.

*For correspondence:
James.boardman@ed.ac.uk

†These authors contributed equally to this work

Competing interests: The authors declare that no competing interests exist.

## Introduction

Prenatal exposure to maternal stress is estimated to affect 10–35% of children worldwide, which is a major concern because early life stress may contribute to impaired cognitive development, negative affectivity, autism spectrum disorder (ASD), and psychiatric diagnoses including attention deficit hyperactivity disorder (ADHD), addiction, depression, and schizophrenia (*Van den Bergh et al., 2017*). Neural correlates of prenatally stressed children include alternations in brain structural and functional connectivity, especially in networks involving the amygdala and prefrontal cortex (*Scheinost et al., 2017*).

Adaptation of the maternal hypothalamic-pituitary-adrenal (HPA) axis is a key mechanism by which maternal stress modulates offspring neurodevelopment (*Moisiadis and Matthews, 2014*), and there is evidence that this mechanism operates in a sexually dimorphic manner (*Sutherland and Brunwasser, 2018*). For example, higher waking maternal salivary cortisol in pregnancy is associated with increased internalising behaviours in female infants and reduced internalising behaviours in

**eLife digest** Stress during pregnancy, for example because of mental or physical disorders, can have long-term effects on child development. Epidemiological studies have shown that individuals exposed to stress in the womb are at higher risk of developmental and mood conditions, such as ADHD and depression. This effect is different between the sexes, and the biological mechanisms that underpin these observations are poorly understood.

One possibility is that a baby's developing amygdala, the part of the brain that processes emotions, is affected by a signal known as cortisol. This hormone is best known for its role in coordinating the stress response, but it also directs the growth of a fetus. Tracking fetal amygdala changes as well as cortisol levels in the pregnant individual could explain how stress during pregnancy affects development.

To investigate, Stoye et al. recruited nearly 80 volunteers and their newborn children. MRI scans were used to examine the structure of the amygdala, and how it is connected to other parts of the brain. In parallel, the amount of cortisol was measured in hair samples collected from the volunteers around the time of birth, which reflects stress levels during the final three months of pregnancy.

Linking the brain imaging results to the volunteers' cortisol levels showed that being exposed to higher cortisol levels in the womb affected babies in different ways based on their sex: boys showed alterations in the fine structure of their amygdala, while girls displayed changes in the way that brain region connected to other neural networks.

The work by Stoye et al. potentially reveals a biological mechanism by which early exposure to stress could affect brain development differently between the sexes, potentially informing real-world interventions.

males (*Braithwaite et al., 2017a*; *Braithwaite et al., 2017b*). Higher maternal salivary cortisol in pregnancy is also associated with stronger amygdala functional connectivity with networks involved in sensory processing and integration in newborn girls, with weaker connectivity to these brain regions in boys (*Graham et al., 2019*); and in childhood, with larger amygdalae (*Buss et al., 2012*) and reduced segregation of structural networks in girls but not boys (*Kim et al., 2016*). The amygdala is further implicated as a neural target of prenatal stress exposure by observations from studies that have characterised maternal stress by symptomatology of depression and/or anxiety, which report alterations in amygdala volume (*Wen et al., 2017*), microstructure (*Rifkin-Graboi et al., 2013*), and functional and structural connectivity among offspring (*Posner et al., 2016*).

Candidacy of the amygdala as an important neural target of prenatal stress exposure comes from the following observations in pre-clinical and clinical studies. First, the amygdala develops early in embryonic life (*Humphrey, 1968*) and contains a high concentration of glucocorticoid receptors (*Wang et al., 2014*); second, increased maternal glucocorticoids modulate amygdala development and anxiety-like behaviours in experimental models (*Welberg et al., 2000*; *Barbazanges et al., 1996*); third, lesion studies in non-human primates support its critical role in early development of emotion regulation (*Schumann et al., 2011*); fourth, newborn amygdala functional connectivity is consistently linked with internalising behaviours in children up to the age of 2 years (*Graham et al., 2019*; *Rogers et al., 2017*); fifth, early disruption to cell composition of the amygdala is reported in a model of early life stress (*Kraszpulski et al., 2006*), and in children with autism (*Avino et al., 2018*); and sixth, in pre-clinical models, stress and glucocorticoid exposure induce dendritic arborisation, amygdala hypertrophy and induce anxiety-like behaviours (*Vyas et al., 2003*; *Mitra and Sapolsky, 2008*).

Neonatal magnetic resonance imaging (MRI) serves as an intermediate phenotype for investigating the impact of early life exposures on brain and health because it is distal to the aetiological process, in this case prenatal stress, and is proximal to cognitive, behavioural and disease outcomes. Structural and diffusion MRI (dMRI) have been used to characterise brain structural maturation and emerging network connectivity during the perinatal period, and to investigate pathways to atypical development (*Batalle et al., 2018*; *Boardman and Counsell, 2020*). It is a suitable tool to investigate the impact of prenatal stress exposure on the amygdala because age-specific templates enable accurate parcellation of the amygdala and associated structures (*Alexander et al., 2017*); and

diffusion tensor imaging and neurite orientation and dispersion density imaging (NODDI) support inference about tissue microstructure and network connectivity, modelled by fractional anisotropy (FA), mean diffusivity (MD), orientation dispersion index (ODI) and neurite density index (NDI) (*Galdi et al., 2020*; *Zhang et al., 2012*).

Hair cortisol concentration (HCC) measured in 3 cm hair samples collected from close to the scalp reflects basal HPA axis activity over the 3 months prior to sampling, and in contrast to single measures from saliva or blood, it is not influenced by short-term activation of the HPA axis in response to acute stressors (*Staufenbiel et al., 2013*). Studies in pregnant women have shown HCC to be an efficient method of retrospective assessment of long-term cortisol secretion, and thus long-term HPA axis activity (*Kirschbaum et al., 2009*; *D'Anna-Hernandez et al., 2011*).

Previous studies have reported sex-specific differences between maternal stress and amygdala functional connectivity and behavioural outcomes among children (*Braithwaite et al., 2017a*; *Braithwaite et al., 2017b*; *Graham et al., 2019*; *Buss et al., 2012*; *Kim et al., 2016*), but study designs leave uncertainty about the mechanism linking maternal stress with amygdala development, the potential confounding role of events and environmental exposures during childhood, and the impact of stress on structural connectivity. Resolving these uncertainties is necessary for developing strategies designed to improve socio-emotional development of children born to women who are stressed during pregnancy. Based on studies of the imaging, biochemical, and clinical phenotype of prenatal stress exposure, we hypothesised that higher levels of maternal HPA activity in the final months of pregnancy ascertained from maternal HCC would impact amygdala development and structural connectivity of offspring infants in a sexually-dimorphic manner, and that these effects would be apparent around the time of birth.

## Results

### Participant characteristics

The parents of 102 infants consented to take part. Of these, two preterm infants died before term equivalent age, 12 did not complete the MRI protocol or images were not amenable processing due to movement artefact; one had an incidental structural anomaly detected at MRI; and nine withdrew before MRI scan. This left data from 78 mother-infant dyads for analysis, the maternal and infant characteristics for whom are shown in *Table 1*. The preterm group (n = 36) had the following co-morbidities: early onset sepsis (n = 1); late onset sepsis (n = 5), necrotising enterocolitis (n = 3); histological chorioamnionitis (n = 14); mechanical ventilation during NICU stay (n = 16, median period of ventilation 2 days [IQR, 1 to 5 days]). Maternal hair was sampled at mean 3.5 ± 2.5 days after delivery, and the median HCC concentration was 5.6 pg/mg (0.5–107.1). Maternal HCC was not associated with gestational age (GA) at birth (r = 0.200, p=0.094). HCC did not differ between mothers of male and female infants (p=0.997). MRI was carried out at term-equivalent age: median 41.9 weeks' GA (range 38.6–45.9).

### Amygdala microstructure

In univariate analysis, there were moderately strong correlations between both FA and MD, and GA at birth and age at scan (r = 0.41–0.64), and weak correlations with birth weight z-score and Scottish Index of Multiple Deprivation 2016 (SIMD2016) quintile (r = 0.24–0.30). There were no significant correlations between FA and MD in amygdalae and ethnicity or infant sex, or maternal parity, age or BMI. There were moderate-to-strong correlations between NDI in the amygdalae and GA at birth and age at scan (r = 0.43–0.74), and a weak correlation with SIMD2016 quintile (r = 0.23–0.26). Weak-to-moderate correlations were observed between ODI in amygdalae with GA at birth and age at scan (r = 0.28–0.42), *Supplementary file 1*.

In multiple linear regression models, there was a significant interaction effect between maternal HCC and infant sex in left amygdala FA (p=0.010) and ODI (p=0.034), with higher maternal HCC being associated with higher left amygdala FA and lower ODI in girls compared to boys (*Table 2*, *Figure 1*). When we stratified by sex, there were associations between maternal HCC and infant amygdala microstructure in boys, but not girls. *Table 3* shows that in boys, higher maternal HCC was associated with lower left amygdala FA (β = −0.339), lower right amygdala FA (β = −0.287) and

**Table 1.** Maternal and neonatal characteristics.

**Maternal characteristics, n = 71**

| | |
|---|---|
| Age (years) | 33.1 ± 5.2 |
| BMI (kg/m$^2$) | 25.2 ± 4.2 |
| Primiparous (%) | 41 (58%) |
| Multiparous (%) | 30 (42%) |
| SIMD 2016 quintile n (%)[*] | |
| 1 | 4 (6%) |
| 2 | 14 (20%) |
| 3 | 10 (14%) |
| 4 | 14 (20%) |
| 5 | 29 (41%) |
| Tobacco smoked during pregnancy, n (%) | 5 (7%) |
| No tobacco smoked during pregnancy, n (%) | 66 (93%) |
| Gestational diabetes (%) | 1 (1%) |
| Preeclampsia (%) | 4 (6%) |
| Receiving pharmacological treatment for depression | 3 (4%) |
| Infant characteristics, n = 78 | |
| Birthweight (g) | 2895 (454–4248) |
| Birth weight z-score[†] | 0.2 ± 1.1 |
| Birth gestation (weeks) | 38.4 (24.0–42.0) |
| Male n (%) | 44 (56%) |
| Female | 34 (44%) |
| European ancestry n (%) | 68 (87%) |
| Other | 10 (13%) |
| Singleton n (%) | 63 (81%) |
| Twin | 15 (19%) |

Normally distributed data is presented as mean ± SD. Non-normally distributed data is presented as median (range).

*Scottish Index of Multiple Deprivation (SIMD) 2016 quintile. First quintile indicates most deprived and fifth quintile the least deprived.

†Calculated according to INTERGROWTH-21st standards.

NDI (β = −0.215), and higher right amygdala MD (β = 0.264) and ODI (β = 0.309), after FDR correction.

**Table 2.** Associations of maternal hair cortisol concentration (HCC) and its interaction with infant sex on amygdala microstructure.

| Side tested | Image metric | Model 1 | Model 2 | | | Model 3 | | |
|---|---|---|---|---|---|---|---|---|
| | | R$^2$ | R$^2$ | HCC β | HCC p-value | R$^2$ | 'HCC x sex' β | 'HCC x sex' p-value |
| Left | FA | 0.267 | 0.269 | −0.048 | 0.858 | 0.359 | 0.677 | 0.010 |
| | MD | 0.405 | 0.405 | 0.018 | 0.858 | 0.413 | −0.191 | 0.358 |
| | ODI | 0.105 | 0.111 | 0.082 | 0.858 | 0.181 | −0.597 | 0.034 |
| | NDI | 0.530 | 0.534 | −0.064 | 0.858 | 0.561 | 0.375 | 0.052 |
| Right | FA | 0.289 | 0.308 | −0.148 | 0.269 | 0.342 | 0.415 | 0.083 |
| | MD | 0.492 | 0.523 | 0.189 | 0.143 | 0.530 | −0.183 | 0.326 |
| | ODI | 0.214 | 0.228 | 0.128 | 0.269 | 0.283 | −0.527 | 0.083 |
| | NDI | 0.554 | 0.562 | −0.094 | 0.269 | 0.583 | 0.330 | 0.083 |

Model 1: Age at MRI, gestational age at birth, birth weight z-score, Scottish Index of Multiple Deprivation 2016 quintile, infant sex. Model 2: Model 1 + (maternal HCC). Model 3: Model 2 + (maternal HCC x infant sex interaction). p-values are FDR adjusted. FA, fractional anisotropy; MD, mean diffusivity; ODI, orientation dispersion index; NDI, neurite density index; HCC, hair cortisol concentration, R$^2$, coefficient of determination; β, standardised beta coefficient; p-value, FDR adjusted probability value.

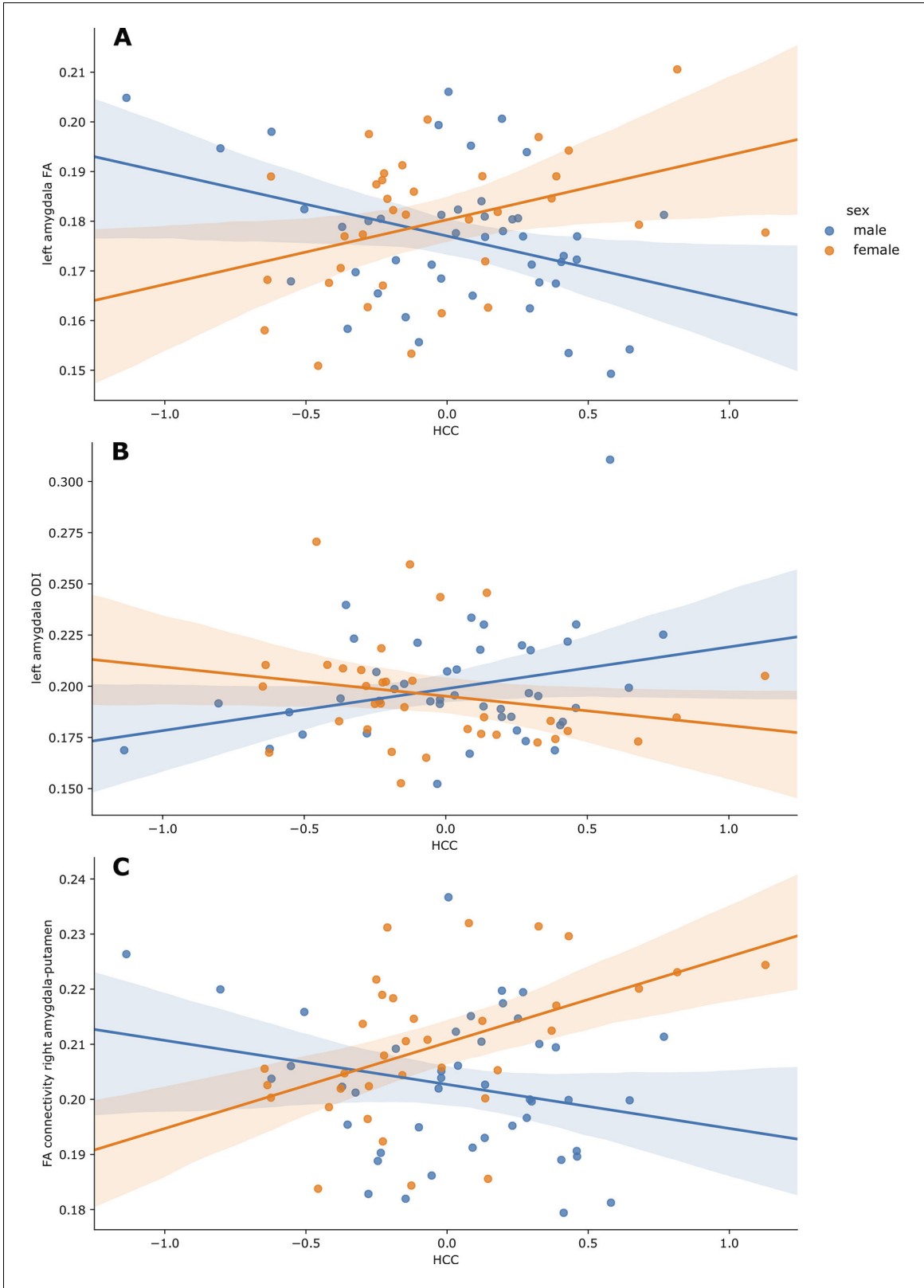

**Figure 1.** Partial regression plots of HCC and (**A**) left amygdala FA, (**B**) left amygdala ODI and (**C**) FA of connections between the right amygdala and putamen. Plots are residualised for GA at birth, GA at scan, birthweight z-score and SIMD quintile. Boys are shown in blue, and girls in orange, along with the 95% confidence intervals.

*Figure 1 continued on next page*

*Figure 1 continued*

The online version of this article includes the following source data for figure 1:

**Source data 1.** Demographic and information and image metrics.

## Structural connectivity of the amygdala

For both hemispheres, the networks with the top 20% number of streamlines were connected to eight structures: thalamus, putamen, insula, superior temporal gyrus, inferior temporal gyrus, middle temporal gyrus, caudate, and lateral orbitofrontal cortex (*Figure 2* and *Figure 2—figure supplement 1*). Quantification of streamline counts is given in *Supplementary file 2* and illustrated in *Figure 3*. Maternal HCC was not associated with streamline counts of the left and right amygdala with these regions.

In fully adjusted analyses, the interaction between maternal HCC and infant sex was significant for mean FA of connections between the right amygdala and putamen. Higher maternal HCC was associated with higher FA for amygdala-putamen connectivity in girls compared with boys (p=0.007) (*Figure 1*). The interaction was also seen for connections to left thalamus, putamen, and insula, but the interaction term did not remain after correction for multiple tests (*Supplementary file 3*). In sex-stratified analysis, girls had higher FA values in association with high maternal HCC in connections between left amygdala with thalamus, putamen, and inferior temporal gyrus, and the right amygdala with putamen and inferior temporal gyrus, but these were not significant after correction for multiple tests (*Supplementary file 3a*).

## Amygdala volume

Mean volumes of the left and right amygdala were $877 \pm 111$ mm$^3$ and $823 \pm 91$ mm$^3$, respectively. In univariate analysis, there were weak associations (r = 0.24–0.3) between amygdala volume and GA at birth and birth weight z-score, but not with age at scan, SIMD2016 quintile, sex, ethnicity, or maternal BMI, parity or age (*Supplementary file 1*). Maternal HCC was not associated with infant right or left amygdala volume in regression models adjusted for potential covariates, and interaction terms between maternal HCC and infant sex were not significant (*Supplementary file 3b*).

## Sensitivity and subgroup analyses

There were seven twin sets in the whole sample. When we repeated analyses including only singletons and the first born of twin pairs, significant associations between maternal HCC, sex and image feature remained, with little change to the value of regression coefficients (*Supplementary file 3c*).

**Table 3.** Associations of maternal hair cortisol concentration (HCC) with amygdala microstructural parameters assessed separately in boys and girls.

| Side tested | Image metric | Boys | | | | Girls | | | |
| | | Model 1 | Model 2 | | | Model 1 | Model 2 | | |
| | | $R^2$ | $R^2$ | HCC β | HCC p-value | $R^2$ | $R^2$ | HCC β | HCC p-value |
|---|---|---|---|---|---|---|---|---|---|
| Left | FA | 0.433 | 0.537 | −0.339 | 0.023 | 0.157 | 0.239 | 0.340 | 0.372 |
| | MD | 0.489 | 0.497 | 0.090 | 0.462 | 0.445 | 0.449 | −0.072 | 0.667 |
| | ODI | 0.124 | 0.215 | 0.317 | 0.085 | 0.132 | 0.159 | −0.194 | 0.648 |
| | NDI | 0.622 | 0.654 | −0.189 | 0.089 | 0.522 | 0.530 | 0.109 | 0.648 |
| Right | FA | 0.368 | 0.443 | −0.287 | 0.047 | 0.301 | 0.306 | 0.091 | 0.736 |
| | MD | 0.508 | 0.571 | 0.264 | 0.047 | 0.497 | 0.506 | 0.111 | 0.736 |
| | ODI | 0.149 | 0.236 | 0.309 | 0.047 | 0.362 | 0.378 | −0.149 | 0.736 |
| | NDI | 0.581 | 0.623 | −0.215 | 0.047 | 0.571 | 0.573 | 0.050 | 0.736 |

Model 1: Gestational age at MRI, gestational age at birth, birth weight z-score, Scottish Index of Multiple Deprivation 2016 quintile. Model 2: Model 1 + (maternal HCC). p-values are FDR adjusted. FA, fractional anisotropy; MD, mean diffusivity; ODI, orientation dispersion index; NDI, neurite density index; HCC, hair cortisol concentration; $R^2$, coefficient of determination; β, standardised beta coefficient; p-value, FDR adjusted probability value.

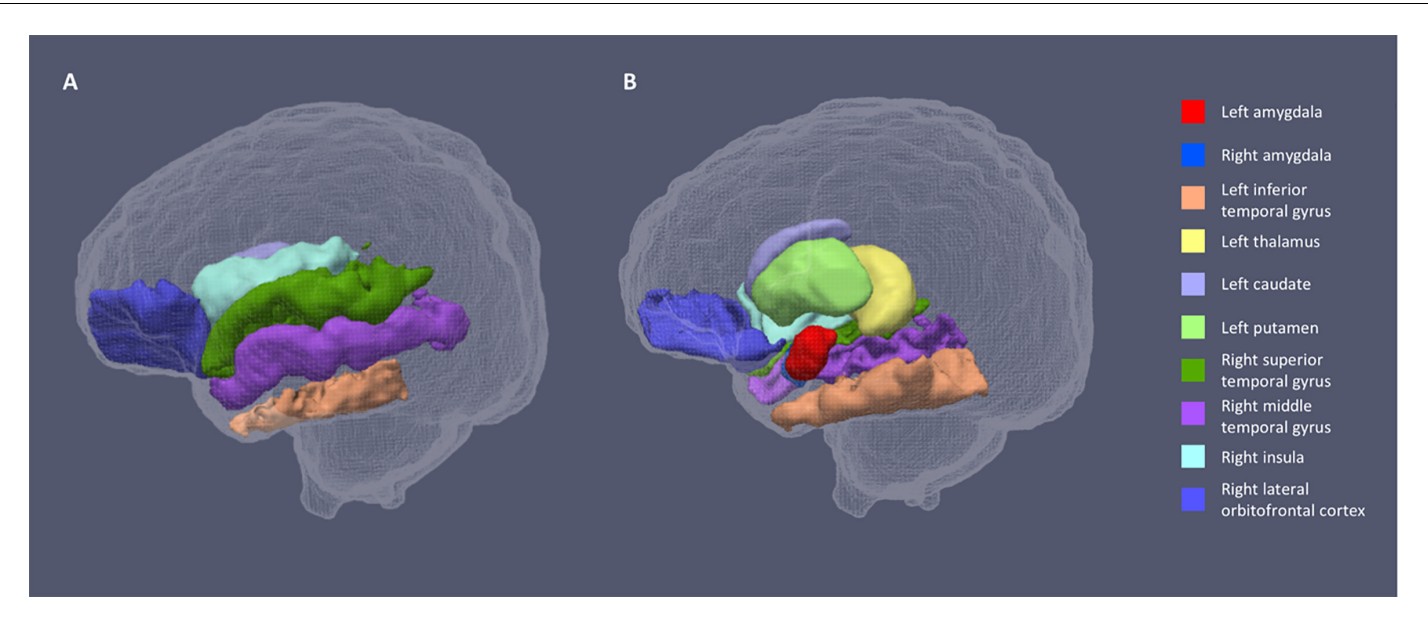

**Figure 2.** Segmentations of the amygdalae and connected regions defined by the top 20% streamline counts. *Figure 1a* shows the lateral view of the sagittal plane and 1b the medial view. The same eight regions had the highest streamline counts to the amygdalae bilaterally.

The online version of this article includes the following figure supplement(s) for figure 2:

**Figure supplement 1.** Example segmentations of the amygdala in native space.

In subgroup analysis of preterm and term infants, the direction and magnitude of interaction effects for both groups were similar to those of the whole sample. Specifically, when tested in term and preterm infants, respectively, higher maternal hair cortisol concentration was associated with higher left amygdala fractional anisotropy ($\beta$ = 0.735 and 0.640), lower left amygdala orientation dispersion index ($\beta$ = −0.710 and −0.614), and higher fractional anisotropy in connections between the right amygdala and putamen ($\beta$ = 0.733 and 0.426) in girls compared to boys (*Supplementary file 3c*).

Bronchopulmonary dysplasia, sepsis, and histological chorioamnionitis did not correlate with any image feature tested. The need for mechanical ventilation did correlate with lower FA in the right amygdala and its connections to putamen, but it had little influence on the interactive effect of sex and maternal cortisol on amygdala microstructure or connectivity (*Supplementary file 3d*).

## Discussion

We report a mechanism that could explain the impact of maternal stress on infant brain development. We found that maternal HCC, a stable marker of chronic maternal HPA axis activity in pregnancy, is associated with microstructure and structural connectivity of the newborn amygdala, a region of functional importance for early social development and emotion regulation. Specifically, HCC interacts with infant sex to modify amygdala FA, ODI, and NDI, which supports the inference that maternal chronic HPA activity has an impact on dendritic structure, axonal configuration, and the packing density of neurites, in a sexually dimorphic manner (*Jespersen et al., 2012*; *Grussu et al., 2017*; *Sato et al., 2017*; *Nazeri et al., 2020*).

The findings are consistent with recent reports from the GUSTO (Growing Up in Singapore Towards Health Outcomes) cohort that describe associations between maternal depressive symptoms and alterations in offspring amygdala development (*Wen et al., 2017*; *Rifkin-Graboi et al., 2013*). That study highlighted the role of maternal mental health on newborn brain development, and focussed attention on the amygdala. Here, we provide mechanistic insights into the relationship between maternal health and wellbeing and amygdala development by using maternal HCC to characterise chronic HPA activity, and the NODDI model for inference about tissue microstructure. We

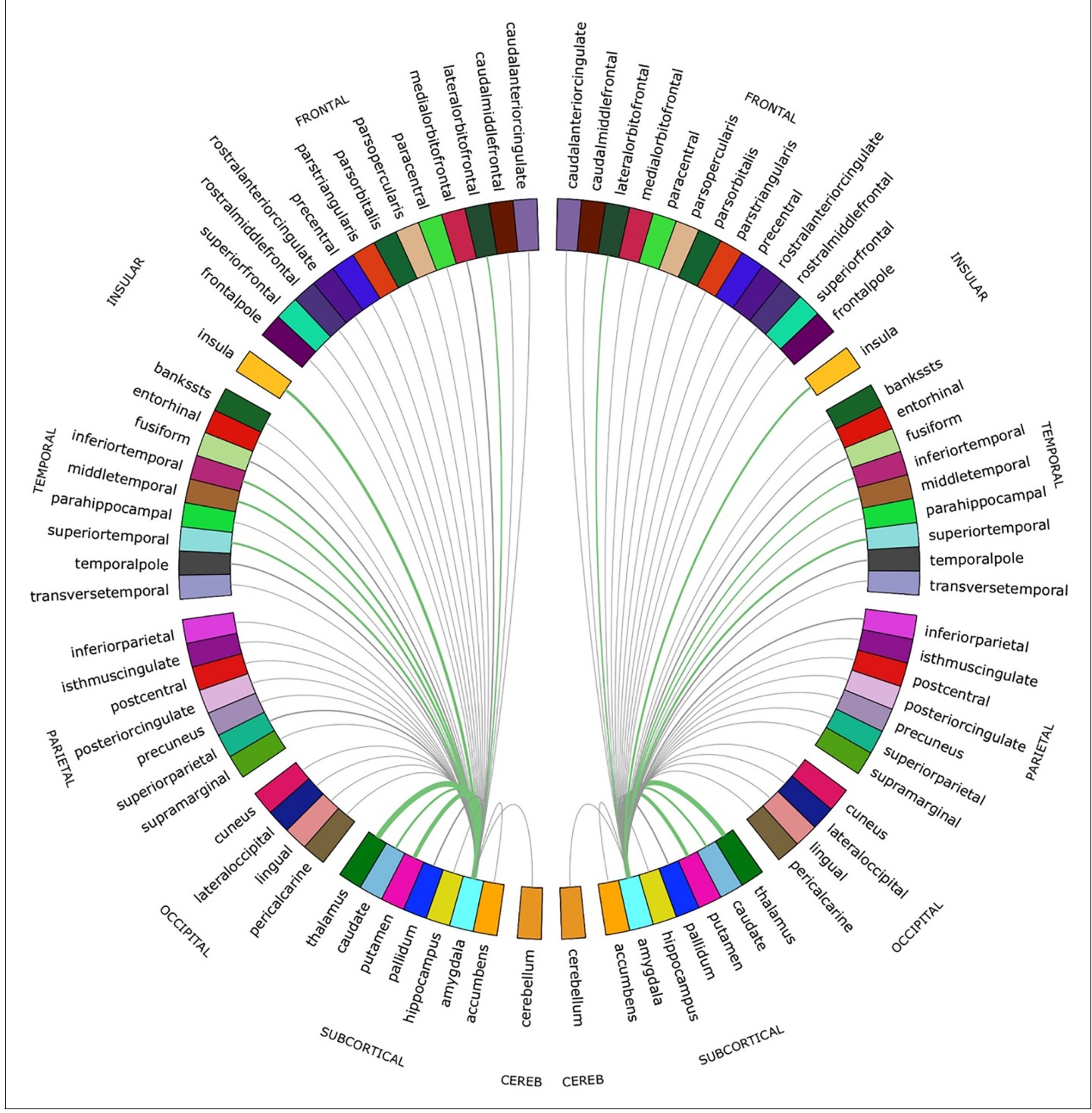

**Figure 3.** Chord diagram of the streamline counts between the amygdalae and unilateral regions of interest (ROIs). The number of streamlines between ROIs are demonstrated by the corresponding arcs thickness. ROIs connected by the top 20% of streamlines are shown in green.

chose to measure NODDI parameters for assessing microstructure because ODI and NDI in grey matter appear to be functionally tractable. For example, diffusion markers of dendritic density and arborisation in grey matter predict differences in intelligence (*Genç et al., 2018*), reduced ODI in grey matter is reported in psychosis and in neurodegenerative disease, and reduced grey matter

NDI is reported in Parkinson's disease, Alzheimer's disease, autism spectrum disorder, and temporal lobe epilepsy (for review see *Nazeri et al., 2020*).

Maternal HCC was also related to structural connectivity of the amygdala in a sex-discordant manner. Higher maternal HCC was associated with higher FA in girls than boys in tracts between right amygdala and putamen. These observations were not explained by differences in streamline counts in relation to maternal HCC. Furthermore, in sex-stratified analysis, there were consistent trends for girls born to women with higher HCC to have higher mean FA between the left amygdala and left thalamus, putamen and inferior temporal gyrus, and between right amygdala and right putamen and right inferior temporal gyrus, although these did not survive FDR correction. During the neonatal period, higher FA in white matter tracts is typically taken to imply microstructural maturation, through increased axon diameter, density, or myelination. Therefore, increased mean FA demonstrated in connections between the amygdala and putamen, in girls exposed to higher cortisol, could be interpreted as increased maturation of these connections.

Several plausible biological mechanisms could underlie sex differences in associations between HCC and fetal neurodevelopment. First, the capacity of the placenta to regulate the passage of cortisol from mother to fetus differs according to fetal sex, evidenced by sex-discordant expression of enzymes controlling glucocorticoid metabolism within the placenta (*Carpenter et al., 2017*; *Meakin et al., 2017*). Second, evidence from gene expression studies indicate that fetal sex influences the direct actions of cortisol in the placenta (*Rosenfeld, 2015*). Third, the same level of cortisol exposure likely holds sex-different actions at the level of the fetal brain, given that male and female fetuses have different glucocorticoid and mineralocorticoid expression across development (*Owen and Matthews, 2003*). Finally, there may be sex differences in placental release of corticotropin-releasing hormone (CRH), which influence fetal cortisol exposure and neurodevelopment (*Bangasser and Wiersielis, 2018*; *Sandman et al., 2013*).

To our knowledge, this is the first study to investigate a physiological measure of chronic maternal HPA activity with quantitative biomarkers of brain development, and to include infants born very preterm. The relationships we describe appear to apply across the whole GA range because GA at birth was included as a covariate in regression models, and in sub-group analyses the magnitude and direction of 'HCC x sex' interaction effects existed in term and preterm groups. This suggests that maternal health and wellbeing across gestation influences neurodevelopment in preterm and term infants. This is important because preterm birth has previously been associated with both exposure to maternal HPA axis dysregulation (*Duthie and Reynolds, 2013*), and an increased risk of inattention and affective disorders (*Johnson, 2007*). Furthermore, the finding that maternal HCC is associated with neonatal amygdala development in preterm and term infants is consistent with the observation that maternal cortisol in early gestation predicts amygdala volume at the age of 7 years (*Buss et al., 2012*). Given that mothers' HPA axis shows considerable trait-stability across pregnancy, whereby women with higher cortisol concentrations in the second trimester also tend to have higher cortisol in the third trimester (*Graham et al., 2019*; *Stoye et al., 2020*), the data lend further support to strategies designed to optimise pre- and early pregnancy health for optimising fetal neurodevelopment.

Strengths of this study are the use of biophysical tissue modelling (NODDI) to enable inference about neurite density and organisation in the amygdala; and use of a data-driven approach to investigate amygdala structural connectivity. A second strength is use of maternal HCC to operationalise stress because it is a quantitative stable marker of cortisol secretion that represents HPA activity over 3 months; as such HCC is unlikely to reflect transient stresses that can occur in pregnancy, and it overcomes the problems of diurnal variation that occur with plasma and saliva measurements.

The study has some limitations: first it was not powered to detect both sex and birth gestation interactions, but this should be considered in future study design. Second, follow-up studies that include measures of socio-emotional development are needed to understand functional consequences of these findings. Finally, the newborn amygdalae are relatively small anatomical regions so could be susceptible to partial volume effects influencing microstructural characteristics. To mitigate this risk, we used an age-specific atlas for segmentation, and excluded voxels with a $\upsilon$iso < 0.5. Third, due to the close proximity of the amygdala and putamen the corresponding connectivity FA may to some degree reflect microstructure of the regions themselves.

Longitudinal studies that evaluate socio-emotional development are needed to understand the functional consequences of these findings. In addition, as the network architecture underlying socio-

emotional function in early life becomes more certain with technological advances, it may be useful to focus image analyses on individual networks. The subcortical-gaze pathway is of specific interest because in adults there is evidence that structural connectivity of the superior colliculus and amygdala is related to processing of facial expressions (*McFadyen, 2019*), which is one of the earliest social cognitive abilities to develop and is foundational to other trajectories of cognitive development.

In conclusion, dMRI and HCC were used to investigate mechanisms underlying the transmission of prenatal stressors on infant development. Maternal HCC in pregnancy is associated with newborn amygdala microstructure and structural connectivity, in a sex-dimorphic manner. These findings reveal that the amygdala, a structure of known importance for child development, is susceptible to variations in the prenatal stress environment, and that cortisol imparts sex-specific effects on human fetal neurodevelopment.

# Materials and methods

**Key resources table**

| Reagent type (species) or resource | Designation | Source or reference | Identifiers | Additional information |
|---|---|---|---|---|
| Software, algorithm | Marchenko-Pastur-PCA-based algorithm | PMID:27523449 | | See Materials and methods, section Image Pre-processing |
| Software, algorithm | Minimal processing pipeline of the developing human connectome project (dHCP) | PMID:29409960 | | See Materials and methods, section Image Pre-processing |
| Software, algorithm | Eddy current, head movement and EPI geometric distortions correction with outlier replacement and slice-to-volume registration | DW-MRI registration in FSL | RRID:SCR_009461 | See Materials and methods, section Image Pre-processing |
| Software, algorithm | Bias field inhomogeneity correction | PMID:20378467 | | See Materials and methods, section Image Pre-processing |
| Software, algorithm | Boundary-based registration | PMID:19573611 | | See Materials and methods, section Image Pre-processing |
| Software, algorithm | Affine and symmetric normalisation (SyN) part of ANTS - Advanced Normalisation ToolS | PMID:17659998 | RRID:SCR_0047 57105105105 | See Materials and methods, section Tissue Segmentation and Parcellation |
| Software, algorithm | Joint label fusion | PMID:22732662 | | See Materials and methods, section Tissue Segmentation and Parcellation |
| Software, algorithm | NODDI | NODDI Matlab Toolbox | RRID:SCR_006826 | See Materials and methods, section Microstructure and volumetric assessments |

*Continued on next page*

*Continued*

| Reagent type (species) or resource | Designation | Source or reference | Identifiers | Additional information |
|---|---|---|---|---|
| Software, algorithm | Constrained spherical deconvolution (CSD); MRtrix | PMID:31473352 | RRID:SCR_006971 | See Materials and methods, section Network Construction Analysis |
| Software, algorithm | • Anatomically-constrained tractography (ACT) | PMID:22705374 | | See Materials and methods, section Network Construction Analysis |
| Software, algorithm | Fiber orientation distribution (FOD) | PMID:25109526 | | See Materials and methods, section Network Construction Analysis |
| Software, algorithm | Spherical-deconvolution informed filtering of tractograms two | PMID:26163802 | | See Materials and methods, section Network Construction Analysis |
| Software, algorithm | SPSS | SPSS | RRID:SCR_002865 | |
| Software, algorithm | Seaborn | Python data visualisation library https://seaborn.pydata.org/ http://doi.org/10.5281/zenodo.883859 | RRID:SCR_018132 | *Figure 1* |
| Software, algorithm | ParaView | ParaView | RRID:SCR_002516 | *Figure 2* |
| Software, algorithm | ITK-SNAP | ITK-SNAP | RRID:SCR_002010 | *Figure 2* |
| Software, algorithm | Circos | Circos | RRID:SCR_011798 | *Figure 3* |
| Other | Liquid chromatography-tandem mass spectrometry (LC-MS/MS) | PMID:23584040 | | See Materials and methods, section Maternal hair cortisol concentrations (HCC) |

## Participants

The 'Stress Response Systems in Mothers and Infants' cohort recruited mother-infant dyads from the Royal Infirmary, Edinburgh, between March 2018 and August 2019. It prospectively tests associations of perinatal glucocorticoid exposure with brain development, and early life exposures including preterm birth with infant HPA axis regulation. We recruited mother-infant dyads who presented to NHS services with threatened preterm labour and delivered at ≤32 completed weeks of gestation, and women who delivered ≥37 weeks' gestation and received routine care with their infant on postnatal wards. No participants were recruited from high-risk antenatal clinics. Exclusion criteria were congenital fetal abnormality, chromosomal abnormality or regular maternal corticosteroid use. All women gave written informed consent. Ethical approval was granted by South East Scotland 01 Regional Ethics Committee (18/SS/0006).

## Maternal hair cortisol concentrations (HCC)

Maternal hair was sampled within 10 days of delivery. Hair was cut close to the scalp, at the posterior vertex, and stored in aluminium foil at −20℃. The proximal 3 cm of hair were analysed by liquid chromatography-tandem mass spectrometry (LC-MS/MS), at Dresden Lab Service GmbH (Dresden, Germany), using an established protocol (*Gao et al., 2013*). Adult hair commonly grows at 1 cm/

month (*Wennig, 2000*) and thus hair segments represented maternal HPA axis activity over the last 3 months of pregnancy.

## Demographic and clinical information

Participant demographic information was collected through maternal questionnaire and review of medical records. Collected maternal information included: age at delivery (years), parity (primiparous/multiparous), clinical diagnosis of gestational diabetes, pre-eclampsia, pharmacological treatment for depression during pregnancy, antenatal corticosteroid exposure for threatened preterm birth; body mass index (BMI) calculated at antenatal booking; smoking status defined as having smoked any tobacco in pregnancy; SIMD 2016 quintile rank, a score generated by the Scottish government which measures localities' deprivation according to local income, employment, health, education, geographic access to services, crime and housing. Infant demographics included whether participants were a singleton or twin, ethnicity, GA at birth (weeks), and birth weight z-score calculated according to intergrowth standards (*Villar et al., 2014*).

## Magnetic resonance imaging

### Image acquisition

Infants underwent MRI at term-equivalent age, at the Edinburgh Imaging Facility, RIE. Infants were fed, wrapped and allowed to sleep naturally in the scanner. Flexible earplugs and neonatal earmuffs (MiniMuffs, Natus) were used for acoustic protection. Scans were supervised by a doctor, or nurse trained in neonatal resuscitation.

A Siemens MAGNETOM Prisma 3 T MRI clinical scanner (Siemens Healthcare Erlangen, Germany) and 16-channel phased-array paediatric head and neck coil were used for acquisition (*Boardman et al., 2020*). In brief, we acquired 3D T1-weighted MPRAGE (T1w) (acquired voxel size = 1 mm isotropic) with TI 1100 ms, TE 4.69 ms and TR 1970 ms; 3D T2-weighted SPACE (T2w) (voxel size = 1 mm isotropic) with TE 409 ms and TR 3200 ms; and axial dMRI. dMRI was acquired in two separate acquisitions to reduce the time needed to re-acquire any data lost to motion artefact: the first acquisition consisted of 8 baseline volumes (b = 0 s/mm$^2$ [b0]) and 64 volumes with b = 750 s/mm$^2$, the second consisted of 8 b0, three volumes with b = 200 s/mm$^2$, six volumes with b = 500 s/mm$^2$ and 64 volumes with b = 2500 s/mm$^2$; an optimal angular coverage for the sampling scheme was applied (*Caruyer et al., 2013*). In addition, an acquisition of 3 b0 volumes with an inverse phase encoding direction was performed. All dMRI images were acquired using single-shot spin-echo echo planar imaging (EPI) with 2-fold simultaneous multislice and twofold in-plane parallel imaging acceleration and 2 mm isotropic voxels; except where stated above, all three diusion acquisitions had the same parameters (TR/TE 3400/78.0 ms).

Conventional images were reported by an experienced paediatric radiologist (A.J.Q.) using a structured system (*Woodward et al., 2006*). Images with focal parenchymal injury (defined as post-haemorrhagic ventricular dilatation, porencephalic cyst or cystic periventricular leukomalacia) were not included in the final sample.

## Image pre-processing

Diffusion MRI processing was performed as follows: for each subject the two dMRI acquisitions were first concatenated and then denoised using a Marchenko-Pastur-PCA-based algorithm (*Veraart et al., 2016*); the eddy current, head movement and EPI geometric distortions were corrected using outlier replacement and slice-to-volume registration (*Smith et al., 2004*; *Andersson et al., 2003*; *Andersson and Sotiropoulos, 2016*; *Andersson et al., 2017*); bias field inhomogeneity correction was performed by calculating the bias field of the mean b0 volume and applying the correction to all the volumes (*Tustison et al., 2010*).

The T2w images were processed using the minimal processing pipeline of the developing human connectome project (dHCP) to obtain the bias field corrected T2w, the brain masks and the different tissue probability maps (*Makropoulos et al., 2018*). The mean b0 EPI volume of each subject was co-registered to their structural T2w volume using boundary-based registration (*Greve and Fischl, 2009*).

## Tissue segmentation and parcellation

The 10 manually labelled subjects of the M-CRIB atlas (*Alexander et al., 2017*) were registered to the bias field corrected T2w using rigid, affine and symmetric normalisation (SyN) (*Avants et al., 2008*). Next, the registered labels of the 10 atlases were merged using joint label fusion (*Wang et al., 2013*), resulting in a parcellation containing 84 regions of interest (ROIs).

## Microstructure and volumetric assessments

Volumes were calculated from ROIs derived in the structural images. ROIs were propagated to the diffusion native space using the previously computed transformation.

To calculate the tensor derived metric, only the first shell was used. NODDI metrics were calculated using the recommended values for neonatal grey-matter of the parallel intrinsic diffusivity (1.25 $\mu m^2 \cdot ms^{-1}$) (*Zhang et al., 2012*; *Guerrero et al., 2019*). The obtained metrics are: neurite density index (NDI), isotropic volume fraction (υiso) and orientation dispersion index (ODI). The mean FA, MD, ODI, and NDI were calculated for the left and right amygdalae M-CRIB ROIs, after exclusion of voxels with a υiso < 0.5. Voxels with a υiso < 0.5 were excluded, in order to minimise partial volume effects (*Batalle et al., 2019*).

## Network construction and analysis

Tractography was performed using constrained spherical deconvolution (CSD) and anatomically constrained tractography (*Tournier et al., 2019*; *Smith et al., 2012*) The required 5-tissue type file, was generated by combining the tissue probability maps obtained from the dHCP pipeline with the subcortical structures derived from the parcellation process. Multi-tissue response function was calculated, with a FA threshold of 0.1. The average response functions were calculated. Then, the multi-tissue fiber orientation distribution (FOD) was calculated (*Jeurissen et al., 2014*), and global intensity normalisation on the FODs images was performed. Finally, the tractogram was created, generating 10 million streamlines, with a minimum length of 20 mm and a maximum of 200 mm and a cut-off of 0.05 (default), using backtrack and a dynamic seeding (*Smith et al., 2015a*). To be able to quantitatively assess connectivity, spherical-deconvolution informed filtering of tractograms two (SIFT2) was applied to the resulting tractograms (*Smith et al., 2015a*). The connectivity matrix was constructed using a robust approach, a 2 mm radial search at the end of the streamline was performed to allow the tracts to reach the GM parcellation (*Smith et al., 2015b*). The final connectivity matrices were multiplied by the μ coefficient obtained during the SIFT2 process.

These connectomes gave a quantification of the SIFT2 weights (referred to as the streamline counts), and the mean FA of connections, between both the left and right amygdala to 41 unilateral regions of interest defined through M-CRIB parcellation. In order to focus analysis on to amygdala's most structurally connected areas, these 82 ROIs were thresholded according to the number of streamlines connecting them to the left or right amygdala, with the top 20% (N = 16) of connections taken forward for further analysis testing relationships with maternal HCC.

## Statistical analysis

Analyses were performed using IBM SPSS Statistics Version 25 Armonk, NY: IBM Corp. Continuous data are summarised as mean ± SD if they had a normal distribution, and median (range) if skewed. Maternal HCC was positively skewed, and log-10 transformed for analysis. The relationship between maternal HCC with infant characteristics was tested using independent t-test and Pearson's correlation for categorical and continuous variables, respectively. Associations between maternal HCC with (i) left and right amygdala microstructure (FA, MD, NDI, ODI), (ii) structural connectivity (number of streamlines and mean FA of connections), (iii) amygdalae volumes were tested using multiple linear regression. In all models, image feature was the dependent variable and maternal HCC was an independent variable. Covariates included infant sex and clinical or demographic factors that were correlated with either left or right amygdala microstructure or volume using Pearson's correlation. Associations with the following were tested: GA at birth, age at scan, birth weight z-score, SIMD2016 quintile, infant ethnicity, infant sex, and maternal parity, BMI and age. Antenatal corticosteroid treatment for threatened preterm birth was not included as a covariate because it was given to n = 36 (100%) women in the preterm group, was highly correlated with GA at birth (r = 0.958, p<0.001), so its inclusion as a covariate would have introduced multicollinearity in regression

analysis. For descriptive purposes, correlations of infant and maternal factors considered as potential covariates are described as weak if r < 0.3, moderate if r = 0.3–0.7, and strong if r > 0.7.

Sex differences in the relationship between maternal HCC and newborn imaging features were assessed by adding an interaction term between maternal HCC and infant sex in the whole group regression model. If a significant interaction was present, sex stratified analysis was conducted independently in boys and girls. Benjamini and Hochberg false discovery rate (FDR) correction was used to adjust p-values for multiple testing. FDR corrections were conducted separately for assessments of left amygdala microstructure (n = 4), right amygdala microstructure (n = 4), left amygdala connectivity (n = 8) and right amygdala connectivity (n = 8).

One sensitivity analysis was carried out to assess whether associations between maternal HCC and image features might be enhanced by inclusion of twins. We repeated analysis of features with a significant 'HCC x sex' interaction in the whole sample, using only singleton pregnancies and the first-born infant of twin pairs. One sub-group analysis of preterm (GA at birth $\leq$32 weeks) and term infants (GA at birth $\geq$37 weeks) was carried out because the relationship between maternal HCC and infant brain development may be gestation specific.

## Acknowledgements

The work was funded by Theirworld (www.theirworld.org) and was undertaken in the MRC Centre for Reproductive Health, which is funded by MRC Centre Grant (MRC G1002033). RMR acknowledges the support of the British Heart Foundation (RE/18/5/34216). Participants were scanned in the University of Edinburgh Imaging Research MRI Facility at the Royal Infirmary of Edinburgh which was established with funding from The Wellcome Trust, Dunhill Medical Trust, Edinburgh and Lothians Research Foundation, Theirworld, The Muir Maxwell Trust and many other sources; we thank the University's imaging research staff for providing the infant scanning.

## Additional information

### Funding

| Funder | Grant reference number | Author |
|---|---|---|
| Theirworld | | James P Boardman |
| Medical Research Council | MRC G1002033 | James P Boardman |
| British Heart Foundation | RE/18/5/34216 | Rebecca M Reynolds |

The funders had no role in study design, data collection and interpretation, or the decision to submit the work for publication.

### Author contributions

David Q Stoye, Conceptualization, Formal analysis, Investigation, Methodology, Writing - original draft, Writing - review and editing; Manuel Blesa, Formal analysis, Writing - review and editing; Gemma Sullivan, Gillian J Lamb, Gill S Black, Investigation, Writing - review and editing; Paola Galdi, Formal analysis, Methodology, Writing - review and editing; Alan J Quigley, Formal analysis; Michael J Thrippleton, Software, Investigation, Writing - review and editing; Mark E Bastin, Investigation, Methodology, Writing - review and editing; Rebecca M Reynolds, Conceptualization, Formal analysis, Supervision, Funding acquisition, Writing - review and editing; James P Boardman, Conceptualization, Formal analysis, Supervision, Methodology, Writing - review and editing

### Author ORCIDs

David Q Stoye ⓘ http://orcid.org/0000-0002-2444-3573
Rebecca M Reynolds ⓘ https://orcid.org/0000-0001-6226-8270
James P Boardman ⓘ https://orcid.org/0000-0003-3904-8960

### Ethics

Human subjects: All mothers gave written informed consent. Ethical approval was granted by South East Scotland 01 Regional Ethics Committee (18/SS/0006).

### Decision letter and Author response

Decision letter https://doi.org/10.7554/eLife.60729.sa1
Author response https://doi.org/10.7554/eLife.60729.sa2

## Additional files

### Supplementary files

• Supplementary file 1. Univariate analysis of potential covariates with left and right amygdala volume and microstructure.

• Supplementary file 2. Streamline counts between amygdalae and atlas regions.

• Supplementary file 3. Associations of maternal hair cortisol concentration (HCC) and its interaction with offspring sex on mean fractional anisotropy (FA) of amygdala networks with high streamline count. (a) Association between maternal hair cortisol concentration (HCC) and fractional anisotropy (FA) weighted connections of the amygdalae in boys and girls (b) Associations of maternal hair cortisol concentration (HCC) and its interaction with infant sex on amygdalae volume (c) Sensitivity and subgroup analyses assessing hair cortisol concentration (HCC) and infant sex interactions (d) Pearson's correlations between perinatal exposures and amygdala volume, microstructure and connectivity.

• Transparent reporting form

• Reporting standard 1. STROBE Checklist for Observational Studies.

### Data availability

All data generated or analysed during this study are included in the manuscript and supporting files.

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
