## [Decision Letter]

**Acceptance summary:**

The work demonstrates that sex-specific differences in the influence of maternal cortisol, previously shown in childhood and adolescence, are present in the neonatal brain. The use of term age neonates allows the exclusion of the confound of childhood experience and exposure, seen in studies looking at older children. This study can therefore specifically implicate prenatal maternal cortisol exposure as an influence on this early maturing and behaviourally important structure.

**Decision letter after peer review:**

Thank you for submitting your article "Maternal cortisol is associated with neonatal amygdala microstructure and connectivity in a sexually dimorphic manner" for consideration by *eLife*. Your article has been reviewed by three peer reviewers, including Jonathan O’Muicheartaigh as the Reviewing Editor and Reviewer #1, and the evaluation has been overseen by Tamar Makin as the Senior Editor.

The reviewers have discussed the reviews with one another and the Reviewing Editor has drafted this decision to help you prepare a revised submission.

As the editors have judged that your manuscript is of interest, but that, as described below, revisions are required before it is published. We would like to draw your attention to changes in our revision policy that we have made in response to COVID-19 (https://elifesciences.org/articles/57162). First, because many researchers have temporarily lost access to the labs, we will give authors as much time as they need to submit revised manuscripts. We are also offering, if you choose, to post the manuscript to bioRxiv (if it is not already there) along with this decision letter and a formal designation that the manuscript is "in revision at *eLife*". Please let us know if you would like to pursue this option. (If your work is more suitable for medRxiv, you will need to post the preprint yourself, as the mechanisms for us to do so are still in development.)

Summary:

This study investigates the relationship between maternal cortisol levels (measured from hair) and infant amygdala microstructure in a cohort of 78 mother-infant dyads. The neonates were a mix of those born at term (n=42) and those born prematurely (n=36) but all were scanned at term equivalent age. The authors demonstrate sex-stratified relationships, with a strong relationship between cortisol and amygdala microstructure in males and a relationship between amygdala and other temporal/subcortical regions in females.

The reviewers agreed that the manuscript is both interesting and timely. The imaging methods are well performed. However, we shared a series of concerns that should be addressed before we can consider publication.

Essential revisions:

Sample:

– The incidence of preterm birth is ~10%, while in this cohort the incidence is much higher. Were the women recruited from a high-risk pregnancy clinic (which may be a high stress population) or do the gestational ages largely reflect twins included in the sample?

– All infants were imaged at term for this protocol. However, NICU-related procedures occurring between birth and scan (primarily days of mechanical ventilation and infection) are associated with alterations subcortical development in preterms. Was the preterm cohort a critically-ill cohort? Were these clinical variables available?

– Maternal education is an important predictor of brain development and outcome. Had the authors considered to adjust for this variable in their analyses, particularly for the volumetric analyses?

Stress and cortisol:

– The last line of the Abstract implicates that the amygdala cortisol relationship gives an insight into the relationship between maternal stress and child outcome. Cortisol levels are shown here to covary with microstructure, but do they reflect actual maternal stress in pregnancy in this sample? Are there any maternal stress (anxiety or depression) questionnaires that could be reported to address this?

Sex Divergence:

– It's not just one sex that is affected but rather sex-specific effects dependent on the outcome examined (amygdala microstructure in male babies and microstructure of connecting white matter from the amygdala in female babies). The Abstract doesn't describe this divergence, focusing on results only with respect to female neonates but actually the interaction effects both sexes in different ways/regions. It would also be very helpful to have plots to illustrates the relationships (residualised for covariates).

Interpretation of measures:

– The major conclusions concerning HCC, the interaction with infant biological sex and the diffusion measures revealing evidence for alterations in "dendritic structure, axonal configuration, and the packing density of neurites…" aren't entirely supported by the findings based on the results from volumetric analyses. The alterations in ODI seen with HCC should be paralleled by changes in the volume data.

Given that brain volume differences are also believed to underlie alterations in dendritic structure, the authors' conclusions wouldn't entirely be supported. Modifying the central claims would be recommended but more data aren't required to support the findings.

– The lack of a significant association between macrostructural changes in the amygdala and HCC was surprising. This is in consideration of previous work in the area in the GUSTO cohort, which focused primarily on amygdalar volumes. The discussion of the results related to the volume data should be expanded upon.

– How inter-related were the measures – e.g. is there a negative association between amygdala microstructure and amygdala connections that could explain the split in sex associations and directionality.

Discussion:

– What potential biological mechanism do the authors propose underlies these sex specific results. There are only really two vague sentences on this, but the complex results need more.

– The authors present an extremely comprehensive overview of the connectivity of the amygdala. Given the authors' conclusions regarding future social cognition assessments, it's surprising to see that the subcortical gaze pathway was not examined (amygdala, thalamus, superior colliculus) as this pathway rapidly processing eye/face processing. Examining connectivity with midbrain structures might be infeasible in the neonatal brain; however, including a discussion of this pathway would be useful for future research in the area.

– For the HCC relationships with microstructure, maternal HCC values for the preterm infants reflects exposure during (coarsely) a different trimester to the term-born infants. The subgroup analyses (term/preterm) indicate that the results are largely independent of this so does this implicate that the influences on amygdala development are from early gestation?

---

## [Author Response]

Essential revisions:Sample:– The incidence of preterm birth is ~10%, while in this cohort the incidence is much higher. Were the women recruited from a high-risk pregnancy clinic (which may be a high stress population) or do the gestational ages largely reflect twins included in the sample?

The incidence of preterm birth is higher in the study group compared of the general population because the study is designed to investigate the impact of perinatal glucocorticoid exposure on brain development, and perinatal factors (including preterm birth) on infant HPA axis regulation. As such, we recruited term and preterm mother-infant dyads. Preterm dyads were recruited when mothers presented with threatened preterm labour (or immediately after birth if preterm delivery happened without time for antenatal informed consent). The analysis included infants born at term in order to understand generalisability of findings, and because existing literature suggests maternal HPA activity impacts fetal development across the whole of pregnancy. No participating woman was recruited from a high risk or multiple pregnancy clinic.

We have added the following clarification to recruitment section of the Materials and methods:

“We recruited mother-infant dyads who presented to NHS services with threatened preterm labour and delivered at ≤32 completed weeks of gestation, and women who delivered ≥37 weeks’ gestation and received routine care with their infant on postnatal wards. No participants were recruited from high risk antenatal clinics.”

– All infants were imaged at term for this protocol. However, NICU-related procedures occurring between birth and scan (primarily days of mechanical ventilation and infection) are associated with alterations subcortical development in preterms. Was the preterm cohort a critically-ill cohort? Were these clinical variables available?

Thank you. We agree that co-morbidities of preterm birth can affect brain development that is manifest on MRI at term equivalent age. Within the preterm group (n=36) there were the following co-morbidities: early onset sepsis (n=1); late onset sepsis (n=5), necrotizing enterocolitis (n=3); histological chorioamnionitis (n=14); mechanical ventilation during NICU stay (n=16, of which the median period of ventilation was 2 days [IQR 1 to 5 days]).

To investigate whether postnatal co-morbidities influenced associations between maternal cortisol and image metrics we have taken the following steps. First, we tested for univariate associations between the perinatal exposure and image metrics. Where significant univariate associations were identified, we performed additional sensitivity analyses by including the relevant perinatal variable in regression models testing associations between maternal cortisol and image metrics.

Univariate analyses are shown in Author response table 1. Bronchopulmonary dysplasia (BPD), chorioamnionitis and late onset sepsis were not associated with amygdala volume, microstructure or the FA of connections to the putamen. However, infants who received mechanical ventilation (coded as any vs. no ventilation) did have lower right amygdala FA (r=-0.332*, p=0.048) and lower FA in connections with the putamen (r=-0.339**, p=0.043).

Author response table 1: Pearson’s correlations between perinatal exposures and amygdala volume, microstructure and connectivity

Therefore, we included mechanical ventilation as a variable in regression models and found it had little influence on the interactive effect of sex and maternal cortisol on left amygdala FA (β=0.660, previously β=0.677), left amygdala ODI (β=-0.584, previously β=-0.597) and the FA of connections between the right amygdala and putamen (β=0.590, β=0.583).

Since perinatal co-morbidities either did not hold univariate associations with amygdala volume, microstructure or connectivity, or change reported associations between HCC and image metrics, we have not amended the analysis reported in the main manuscript but we have added the following lines to the Results section and we have included the analysis detailed above in Supplementary file 3D:

“The preterm group (n=36) had the following co-morbidities: early onset sepsis (n=1); late onset sepsis (n=5), necrotizing enterocolitis (n=3); histological chorioamnionitis (n=14); mechanical ventilation during NICU stay (n=16, median period of ventilation 2 days [IQR, 1 to 5 days]).”

“Bronchopulmonary dysplasia, sepsis, and histological chorioamnionitis did not correlate with any image feature tested. The need for mechanical ventilation correlated with lower FA in the right amygdala and its connections to putamen, but it had little influence on the interactive effect of sex and maternal cortisol on amygdala microstructure or connectivity (Supplementary file 3D).”

– Maternal education is an important predictor of brain development and outcome. Had the authors considered to adjust for this variable in their analyses, particularly for the volumetric analyses?

Maternal education (binarized as university or college educated (yes [56], no [20]) was not associated with left amygdala (r -0.49) or right amygdala (r -0.148) volume.

Stress and cortisol:– The last line of the Abstract implicates that the amygdala cortisol relationship gives an insight into the relationship between maternal stress and child outcome. Cortisol levels are shown here to covary with microstructure, but do they reflect actual maternal stress in pregnancy in this sample? Are there any maternal stress (anxiety or depression) questionnaires that could be reported to address this?

The literature linking maternal cortisol with psychological stress/psychiatric disease (‘stress’) in pregnancy does not report a consistent relationship between the two. Our own analysis of the ABCD cohort (Reynolds RM), showed that maternal cortisol is mainly associated with biological and lifestyle factors, and not with psychosocial factors (Blekker et al. Psychoneuroendocrinology 2017).

We have clarified this point of terminology by changing the term ‘…stress…’ to ‘…health…’ in the Abstract.

Sex Divergence:– It's not just one sex that is affected but rather sex-specific effects dependent on the outcome examined (amygdala microstructure in male babies and microstructure of connecting white matter from the amygdala in female babies). The Abstract doesn't describe this divergence, focusing on results only with respect to female neonates but actually the interaction effects both sexes in different ways/regions. It would also be very helpful to have plots to illustrates the relationships (residualised for covariates).

Thank you for these helpful suggestions. The following sentences have been added to the Abstract:

“We found a relationship between maternal HCC and amygdala development that differed according to infant sex.”

“…Furthermore, altered amygdala microstructure was only observed in boys, with connectivity changes restricted to girls.”

Residualised plots showing sex-discordant associations between HCC, and left amygdala FA, left amygdala ODI and the FA of connections between the right amygdala and right putamen have been added to the manuscript, as requested (Figure 1).

Interpretation of measures:– The major conclusions concerning HCC, the interaction with infant biological sex and the diffusion measures revealing evidence for alterations in "dendritic structure, axonal configuration, and the packing density of neurites…" aren't entirely supported by the findings based on the results from volumetric analyses. The alterations in ODI seen with HCC should be paralleled by changes in the volume data.Given that brain volume differences are also believed to underlie alterations in dendritic structure, the authors' conclusions wouldn't entirely be supported. Modifying the central claims would be recommended but more data aren't required to support the findings.

The microstructural measures we used (NDI, ODI, FA and MD) are generally considered to support inference about dendritic structure, axonal configuration, and the packing density of neurites (see citations in manuscript). In the neonatal period, it is very common for microstructural alternations to be present without volumetric correlates, and in adulthood, alterations in dMRI parameters in the contexts of disease and cognition and are not necessarily accompanied by changes in macrostructure (Nazari et al., 2020). Therefore, we think there are insufficient data in the literature to be certain that changes in dMRI parameters, including ODI, should necessarily be *paralleled* by changes in volume.

Further considerations are: first, the neonatal brain has a higher water content than at other times in life, and this is likely to influence the relationship between microstructure and macrostructure; and second, it is plausible that maternal factors influence the developmental trajectory of the amygdala (Tottenham and Sheridan, 2010) by affecting microstructure in the newborn period, which might in turn affect network connectivity through development, manifesting as volumetric alterations in later childhood. Please also see the next response, which is related to this point.

– The lack of a significant association between macrostructural changes in the amygdala and HCC was surprising. This is in consideration of previous work in the area in the GUSTO cohort, which focused primarily on amygdalar volumes. The discussion of the results related to the volume data should be expanded upon.

The publications arising from the GUSTO cohort that are directly relevant to our study are listed below. There are consistencies between our study and the GUSTO studies when timing of neuroimaging is taken into account. Associations between maternal depressive symptoms and amygdala volume in the GUSTO cohort were found when MRI was conducted in childhood (Wen et al., 2017). However, when the relationship was assessed in the newborn period, maternal depression was associated with amygdala FA, but not with amygdala volume (Rifkin-Graboi et al., 2013). Furthermore, the GUSTO cohort provides supportive evidence that volumetric deficits in sub-cortical structures (hippocampus) in the context of maternal depression may not be apparent at birth but emerge over time (Qiu et al., 2013).

Although we do not see contradictions between the results reported in this manuscript and those of the GUSTO study when timing of imaging is considered, we question the expectation that the two studies should necessarily yield similar results because there are major differences between study designs. The GUSTO investigators used maternal depressive symptoms as the predictor variable in the analyses, whereas we used maternal cortisol; as noted in the response to the point ‘Stress and cortisol’, this is determined by biological and lifestyle factors and is only inconsistently correlated with mental health problems in pregnant women.

– How inter-related were the measures – e.g. is there a negative association between amygdala microstructure and amygdala connections that could explain the split in sex associations and directionality.

Measures of microstructure and connectivity are correlated with varying degrees of strength (see Author response image 1). It can be seen that correlations between microstructural measures are broadly similar in boys and girls.

**Author response image 1. sa2fig1:** Pearson’s correlations between amygdala microstructural and the putamen to amygdala connectivity, in boys at the bottom left and girls at the top right.

Discussion:– What potential biological mechanism do the authors propose underlies these sex specific results. There are only really two vague sentences on this, but the complex results need more.

Thank you. We have expanded the Discussion to include the following:

“Several plausible biological mechanisms could underlie sex differences in associations between HCC and fetal neurodevelopment. […] Finally, there may be sex differences in placental release of corticotropin-releasing hormone (CRH) which influence fetal cortisol exposure and neurodevelopment (Bangasser and Wiersielis, 2018; Sandman, Glynn and Davis, 2013).”

– The authors present an extremely comprehensive overview of the connectivity of the amygdala. Given the authors' conclusions regarding future social cognition assessments, it's surprising to see that the subcortical gaze pathway was not examined (amygdala, thalamus, superior colliculus) as this pathway rapidly processing eye/face processing. Examining connectivity with midbrain structures might be infeasible in the neonatal brain; however, including a discussion of this pathway would be useful for future research in the area.

We agree this is an interesting future direction to discuss. We have added the following to the Discussion:

“Longitudinal studies that evaluate socio-emotional development are needed to understand the functional consequences of these findings. […] The subcortical-gaze pathway is of specific interest because in adults there is evidence that structural connectivity of the superior colliculus and amygdala is related to processing of facial expressions (McFadyen, 2019), which is one of the earliest social cognitive abilities to develop and is foundational to other trajectories of cognitive development.”

– For the HCC relationships with microstructure, maternal HCC values for the preterm infants reflects exposure during (coarsely) a different trimester to the term-born infants. The subgroup analyses (term/preterm) indicate that the results are largely independent of this so does this implicate that the influences on amygdala development are from early gestation?

Yes, the analysis supports this inference. We have emphasised this point for clarity in the Discussion (fifth paragraph).

References:

Nazeri A et al., in vivo Imaging of Gray Matter Microstructure in Major Psychiatric Disorders: Opportunities for Clinical Translation. Biol Psychiatry Cogn Neurosci Neuroimaging. **5**, 9, 855-864 (2020).

Qiu A, Rifkin-Graboi A, Chen H, Chong YS, Kwek K, Gluckman PD, et al. Maternal anxiety and infants' hippocampal development: timing matters. Transl psychiatry. 2013;3(9):e306-e306.Rifkin-Graboi A et al., Prenatal maternal depression associates with microstructure of right amygdala in neonates at birth. Biological psychiatry **74**, 837-844 (2013).

Tottenham N, Sheridan MA., A review of adversity, the amygdala and the hippocampus: a consideration of developmental timing. Front Hum Neurosci. 3, 68 (2010).

Wen DJ et al., Influences of prenatal and postnatal maternal depression on amygdala volume and microstructure in young children. Transl psychiatry **7**, e1103-e1103 (2017).